# Detection of microplastics in human saphenous vein tissue using µFTIR: A pilot study

**Jeanette M. Rotchell[1] \*, Lauren C. Jenner[2], Emma Chapman[1], Robert T. Bennett[3], Israel Olapeju Bolanle[2], Mahmoud Loubani[3], Laura Sadofsky[2], James Hobkirk[4], Timothy M. Palmer[2]**

**1** Department of Biological and Marine Sciences, University of Hull, Hull, United Kingdom, **2** Centre for Biomedicine, Hull York Medical School, University of Hull, Hull, United Kingdom, **3** Department of Cardiothoracic Surgery, Castle Hill Hospital, Cottingham, United Kingdom, **4** Research Fellow, Department of Cardiology & Cardiothoracic Surgery, Hull University Teaching Hospitals NHS Trust

\* J.Rotchell@hull.ac.uk

**Data Availability Statement:** All relevant data are within the paper and its Supporting Information files. We also set up a figshare file as indicated in the revised manuscript response to reviewers document.

## Abstract

Microplastics (MPs) are ubiquitous in the environment, in the human food chain, and have been recently detected in blood and lung tissues. To undertake a pilot analysis of MP contamination in human vein tissue samples with respect to their presence (if any), levels, and characteristics of any particles identified. This study analysed digested human saphenous vein tissue samples (n = 5) using µFTIR spectroscopy (size limitation of 5 µm) to detect and characterise any MPs present. In total, 20 MP particles consisting of five MP polymer types were identified within 4 of the 5 vein tissue samples with an unadjusted average of 29.28 ± 34.88 MP/g of tissue (expressed as 14.99 ± 17.18 MP/g after background subtraction adjustments). Of the MPs detected in vein samples, five polymer types were identified, of irregular shape (90%), with alkyd resin (45%), poly (vinyl propionate/acetate, PVAc (20%) and nylon-ethylene-vinyl acetate, nylon-EVA, tie layer (20%) the most abundant. While the MP levels within tissue samples were not significantly different than those identified within procedural blanks (which represent airborne contamination at time of sampling), they were comprised of different plastic polymer types. The blanks comprised n = 13 MP particles of four MP polymer types with the most abundant being polytetrafluoroethylene (PTFE), then polypropylene (PP), polyethylene terephthalate (PET) and polyfumaronitrile:styrene (FNS), with a mean ± SD of 10.4 ± 9.21, $p$ = 0.293. This study reports the highest level of contamination control and reports unadjusted values alongside different contamination adjustment techniques. This is the first evidence of MP contamination of human vascular tissues. These results support the phenomenon of transport of MPs within human tissues, specifically blood vessels, and this characterisation of types and levels can now inform realistic conditions for laboratory exposure experiments, with the aim of determining vascular health impacts.

**Funding:** This research did not receive any specific grant from funding agencies in the public, commercial, or not-for-profit sectors. It was part funded by a PhD scholarship for LJ in the "Human Health and Emerging Environmental Contaminants" cluster funded by the University of Hull.

**Competing interests:** The authors have declared that no competing interests exist.

## Introduction

In 2019, pollution was reported as responsible for approximately 9 million premature deaths, with air pollution (both household and ambient air pollution) being responsible for over 70% [1]. There is also a growing body of data that links environmental pollution, particularly $PM_{2.5}$ levels, with poor cardiovascular outcomes [for review: [2]). The precise composition of particulate matter in the $PM_{2.5}$ or $PM_{10}$ category is not known. Microplastics (MPs) are attracting interest as an emerging contaminant of concern that are present in these fractions. MPs are particles of synthetic polymers in the micro size range and while an international consensus on size range has not been reached, the typical range is between 1 μm and 5 mm [3]. MPs have been identified in all environmental compartments [4–6] food, drinking water [7,8], and more recently in human samples including stool [9], blood [10], cadaver lung [11], lung [12] and colon [13]. The biological impacts of MP exposure investigations using human cell and tissue approaches have started to be characterised; with inflammatory and oxidative stress type responses reported (for review: [14]). In a pilot study using blood taken from two healthy individuals, toxic effects of polyethylene (PE) beads, sizes 10–45 μm, with a 48-h exposure of 25–500 μg/mL, were observed to cause increased level of genomic instability, as evidenced by increased micronucleation, nucleoplasmic bridge formation and nuclear bud formation in human peripheral blood lymphocytes [15]. Aside from such cell and tissue culture studies, the clinical implications of MPs within the human body have not yet been determined. Evidence is emerging however, that high MP levels are associated with inflammatory disorders, specifically bowel disease [16].

Saphenous vein autografts are widely used as conduits in coronary artery bypass graft (CABG) procedures aimed at diverting blood around narrowed or blocked coronary arteries to restore blood supply to the heart in patients with coronary heart disease [17]. However, 40–50% of CABG procedures ultimately fail after 10 years due to a variety of factors [18]. Importantly, no published studies have examined any potential link between environmental MP exposure and CABG outcomes. A key first step is, therefore, to assess MP infiltration and composition in vascular conduits. Equally importantly, no published studies have examined whether MPs can infiltrate and or cross any biological barrier, including blood vessels. This study, therefore, aims to identify MP particles present in digested human saphenous vein tissue samples, while also accounting for procedural blank contamination. Any particles isolated from vein tissue have been chemically characterised using μFTIR spectroscopy.

## Methods

### Human tissue acquisition

Excess human saphenous vein tissue was collected, with written consent, from CABG procedures carried out at the Department of Cardiothoracic Surgery, Castle Hill Hospital, Hull University Teaching Hospitals NHS Trust, under NHS Research Ethics Committee and Health Research Authority approval (ref. 15/NE/0138, IRAS project ID 170899). Tissue samples were collected from patients undergoing surgery, between the dates 02/2022 and 03/2022. Details of the donors smoking status, occupation and area of residence were unavailable for the researchers under the terms of the ethical approval obtained. Tissue samples were placed into empty glass containers with foil lids and placed on ice, transported to the laboratory, and digestion conducted on the same day. Vein tissue was obtained from 5 patients, (n = 5, total tissue mass = 4.66 g), resulting in a mean mass of 0.93 ± 0.52 g (range 0.23–1.54 g). Patients mean age was 71.8 ± 2.5 years (range 68–75), two females and three males (Table 1).

**Table 1. Patient and tissue sample information alongside the number of MPs identified within samples by μFTIR spectroscopy.** Polymer types and particle characteristics are included, and three different contamination adjustments are used to display results in units of MP/g of tissue. Abbreviations; resin = alkyd resin, PVAc = poly (vinyl propionate/acetate), PVAE = polyvinyl acetate:Ethylene, TL = Tie Layer consisting of nylon-EVA or ethylene vinyl alcohol (EVOH)-EVA, PUR = polyurethane, PET = polyethylene terephthalate, PTFE = polytetrafluoroethylene, PP = polypropylene, FNS = poly (fumaronitrile:Styrene). [a] shape category unclear.

| ID | Sex | Ethnic origin | Tissue (g) | MP total * | MP polymer | Length, width (µm) | Shape | MP/g † | MP/g †† | MP/g ††† |
|----|-----|---------------|------------|------------|------------|--------------------|-------|--------|---------|----------|
| 1 | M | White | 0.72 | 6 | Resin (4) PVAc (1) PVAE (1) | 236, 75 16,14 97,27 29,11 25,16 19,18 | irregular[a] irregular irregular irregular irregular irregular | 33.33 | 18.89 | 22.22 5.56 5.56 |
| 2 | F | White | 1.51 | 8 | Resin (3) TL (3) PUR (2) | 97,49 300,300 132,30 1074,21 100,59 154,77 27,23 87,86 | irregular irregular irregular fibre irregular irregular irregular irregular | 21.12 | 14.30 | 7.95 7.95 5.23 |
| 3 | M | White | 1.39 | 0 | | | | 0 | 0 | 0 |
| 4 | M | White | 0.81 | 1 | Resin (1) | 36,34 | irregular | 4.94 | 0 | 4.94 |
| 5 | F | White | 0.23 | 5 | Resin (1) PVAc (3) TL (1) | 28,21 21,15 22,15 110,7 20,10 | irregular irregular irregular fibre irregular | 86.96 | 41.74 | 17.39 52.17 4.35 |

Combined MP values: Mean 29.28 ± 34.88 MP/g unadjusted; 14.99 ± 17.18 MP/g with blanks (regardless of polymer type) subtracted

Procedural blank samples

| | | | | | | | | | | |
|----|-----|---------------|------------|------------|------------|--------------------|-------|--------|---------|----------|
| 1 | | | | 2 | PTFE (1) Resin (hydrocarbon) (1) | 171,78 300,50 | fragment fragment | | | |
| 2 | | | | 5 | PET (2) PTFE (1) PP (1) FNS (1) | 20,17 >300,12 48,39 41,24 20,20 | fragment fibre fragment fragment fragment | | | |
| 3 | | | | 0 | | | | | | |
| 4 | | | | 5 | PTFE (3) PP (2) | >300,208 138, 39 151,73 55,54 68,16 | fragment fragment fragment fragment fragment | | | |
| 5 | | | | 1 | PP (1) | 86,50 | fragment | | | |

Combined MP values: Mean 10.40 ± 9.21 MPs.

*On one quarter of a filter, so one quarter of whole sample.

† No account for MPs found in controls, all MPs combined per individual.

†† Total MPs within sample–Total MPs identified in controls (regardless of polymer type).

††† Each MP polymer concentration calculated using LoD/ LoQ method [19] (S1 Table).

## Quality assurance and control measures

Strict control measures were used to quantify and characterise the nature of any unavoidable background contamination as described previously [12]. Due to the ubiquitous nature of MPs in the air and presence of MPs in the surgical environment [20], contamination on the surface

of vein tissue samples could be possible during the surgical procedure, where tissue was removed from live human subjects. Each tissue sample was placed immediately in a glass vial. In parallel to sample processing, 'procedural blanks' (n = 5) were initiated at the same time, during surgery, to quantify and characterise any contamination from the surgical environment and represent contamination from the laboratory reagents, equipment or fallout from the air during the transfer of samples. Procedural blank results were combined to account for contamination at every step. The weight of tissue samples was deduced by taking the weight of the vial plus sample and subtracting the weight of the empty vial to avoid unnecessary additional exposure to any background contamination. Tissue samples were digested individually, alongside a procedural blank, which mimicked the entire tissue processing steps but lacked the vein tissue sample. No standardised protocols are currently adopted within the MPs research field to account for background contamination, so multiple contamination adjustments were applied in this study for comparison (Table 1). Two approaches were used: subtraction, which is routinely used in the MP research field, and a limit of detection (LOD) and limit of quantification (LOQ) approach [19]. Presenting raw data, subtraction, and LOD/LOQ adjusted results allows a comparison for each technique (Table 1).

All reagents were pre-filtered and prepared in bulk. The $H_2O_2$ used was triple filtered using an all-glass vacuum filtration kit and 47 mm glass fibre grade 6 filters (GE Healthcare Life Sciences, Marlborough MA, U.S.A.). All glassware was manually cleaned, dishwashed using distilled water and manually rinsed three times with triple filtered MilliQ water. All equipment and reagents were covered with foil lids and poured through a small opening. After filtering digested samples, glassware and the sides of the filtration kit were rinsed three times with triple filtered MilliQ water to avoid the loss of any sample particles. All work was carried out in a clean fume cupboard with the power off and the shield down to minimise unfiltered air flow [21] and particle suspension [22]. Plastic equipment was avoided, a cotton laboratory coat, and a new set of nitrile gloves for each sample processing step were used. Tissue preparation and particle analysis was conducted at times of low activity in the laboratory and no room ventilation, and µFTIR conducted in a single person room with no windows.

## Vein tissue digestion and filtration

Fresh saphenous vein samples (n = 5) were placed in pre-filtered hydrogen peroxide (200 mL of 30% (v/v) $H_2O_2$) alongside 'procedural blanks' (n = 5). The total mass of each individual tissue sample digested is detailed in Table 1. Flasks were placed in a shaking incubator at 65°C for approximately 7 days, 85 rpm, or until there was no visible tissue. The digest, adapted from previous studies investigating MPs within different environmental and tissue samples [12,23], promotes removal of organic particles while maintaining MP integrity [23]. Samples were then filtered onto aluminium oxide filters (0.02 µm Anodisc, Watford, U.K.) using a glass vacuum filtration system. These were stored in petri dishes before chemical composition analysis alongside blanks.

## Chemical characterisation of particles using µFTIR analysis

Measurements were made using a µFTIR spectrometer (model Nicolet iN10, ThermoFisher, Waltham MA, U.S.A.). Each tissue sample Anodisc filter was placed onto the µFTIR spectroscopy platform, and the length (largest side) and width (second largest side) recorded using the aperture height, width, and angle size selection tool, a visual imaging tool available with the ThermoScientific Omnic Picta Nicolet iN10 microscopy software. Particles were given a shape category (fibre, film, fragment, foam, or sphere [24]), with fibrous particles characterised as having a length to width ratio > 3 [25]. Where the particle shape was ambiguous the term

'irregular' was noted instead and could represent either fragment or film. µFTIR analysis was conducted in liquid nitrogen cooled transmission mode without the aid of further accessories or crystals (Nicolet iN10, ThermoFisher, Waltham MA, U.S.A.). The cooled mercury cadmium telluride (MCT) detector facilitated the analysis of particles accurately down to 5 µm in size. The Nicolet iN10 microscope used is equipped with 15 × 0.7 N.A. high efficiency objective and condenser. It has a colour CCD digital video camera with an independent reflection and transmission illuminations mounted, for capturing images of particles. This model has a standardised 123× magnification with the aperture settings used. No observational criteria was applied to select specific particles for analysis to prevent bias. Using the aperture size selection tool, all particles upon the sample filter >5 µm were included in the analysis process. For this study, a quarter of each filter, containing the total digested tissue sample, was analysed. A background reference spectrum was first recorded for every particle. µFTIR parameters were; spectral range of 4000–1250 cm$^{-1}$, spectral resolution 8 cm$^{-1}$, scan number of 64. Smoothing, baseline correction, spectra normalisation, spectral treatment and data transformation were not used. The manufacturers standard setting for apodization was used. Resulting sample spectra were compared, using the 'Search' option, to a combination of polymer libraries (Omnic Picta, Omnic Polymer Libraries) and full spectral ranges were used with a match threshold of ≥70%. Particles below ≥70% match, and particles not classified as a plastic were recorded but not included in the results shown [26]. The total number of particles (MPs and others) identified was 242, for which 220 (91%) of these were above the 70% hit quality index threshold. Only the MPs data are presented in the results.

## Statistical analysis

Data are presented as mean ± standard deviation. The number of MP particles detected on one quarter of each sample filter and blank filter was multiplied by 4 to determine whole filter/ sample values. These values were then used to calculate the mean ± standard deviation. Tests for homogeneity and significance were performed on unadjusted MP values using SPSS. All data were determined not normally distributed with a Shapiro-Wilk test and a Kruskal-Wallis test applied. There are no standardised methods for the calculation of MP concentrations available at present, herein three are presented: unadjusted, mean of the procedural blank values, regardless of polymer type, subtracted, and an LOD/LOQ method [19].

## Results

### MP abundance levels detected in human vein tissue samples

A total of 20 MP particles were identified (on one quarter of the filters) within 4 of the 5 human saphenous vein tissue samples (Table 1). This extrapolated to (accounting for conversion from one quarter area of a filter analysed, to a whole filter area/sample) an overall unadjusted mean and standard deviation of 16.00 ± 13.56 MPs per sample (range 0–32 MPs) identified within human vein tissue samples, which did not significantly differ ($p$ = 0.293) compared with 10.40 ± 9.21 MPs per sample detected in the combined blanks but did differ in terms of polymer types detected. When considering the mass of the tissue sample, without accounting for background contamination, a mean of 29.28 ± 34.88 MP/g was detected (Table 1). After subtracting background contamination, regardless of polymer type, this value becomes 14.99 ± 17.18 MP/g (Table 1). Both female samples contained at least one MP particle, while one of the three male samples did not. A detailed description of the characterisation of background MP contamination (procedural) is included in Table 1. The MP polymers alkyd resin (Fig 1), poly (vinyl propionate/acetate), PVAc (Fig 1), nylon ethylene vinyl alcohol, EVA,

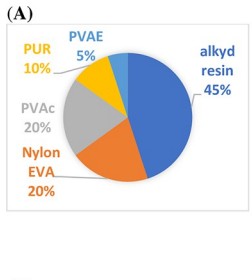

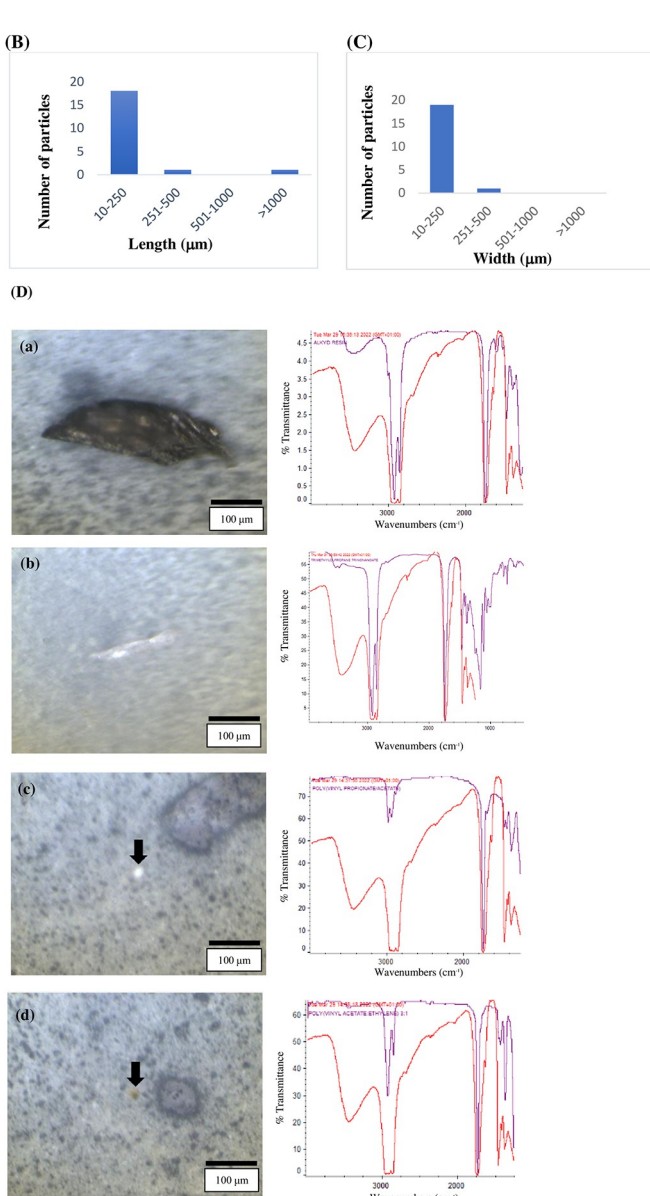

**Fig 1. Microplastic particle characteristics.** Polymer types (A), lengths/widths (B/C), and (D), selected images of the MPs identified within vein tissue samples alongside the spectra obtained (a) alkyd resin fragment, (b) trimethylolpropane trinononoate (plastic additive) containing fragment, (c) poly vinyl propionate/acetate (PVAc) fragment, (d) polyvinyl acetate:Ethylene (PVAE) fragment.

tie layer, and polyurethane, PUR, were above the LOD and LOQ for vein samples from each patient where detected (S1 Table).

## MP particle characterisation from human vein tissue samples

Five MP polymer types were identified in the tissue samples, alkyd resin (9, 45%), PVAc (4, 20%) and nylon-EVA, tie layer (4, 20%) were the most abundant (Fig 1). The majority (90%) of MPs identified within tissue samples were irregular-shaped fragment/film, with an unusual morphological texture (Table 1). MP particles identified within the tissue samples had a mean particle length of 119.59 ± 226.82 μm (range 16–1074 μm), and a mean particle width of 41.27 ± 62.80 μm (range 7–300 μm) (Fig 1).

Several non-MP, yet related, as either MP building block polymer monomers or polymer additives, were also observed as follows: particles comprised of propyleneglycol dilaurate, sorbitan monopalmitate (known as the trade name Span 40), trimethylolpropane trinonanoate, and methyl laurate were detected. Other non-MP particle components included N-methyl-3-piperidylbenzilate HCl, monoelaidin, epoxidized soybean oil, and fatty acid and alcohol ester.

## Characterisation of background MP contamination (procedural blanks)

Particles identified within 'procedural blanks' had a mean MP contamination rate of 10.4 ± 9.21 MP per sample (range 0–20), (Table 1). No particles were identified within 'procedural blank' from the control that arrived alongside vein tissue 3 sample (Table 1). The average length of MPs detected within the combined blank samples was 130.69 ± 107.63 μm (range 20 −>300 μm), and an average width of 52.31 ± 51.42 μm (range 12–208 μm). The shapes of MPs identified in the combined blank samples were all fragments except for one fibre. In addition to MP particles, non-MP natural source polymer particles (such as zein and cellulose) were detected on the sample filters.

## Background MP contamination adjustments

Using adjustments, to account for the procedural blank contamination levels detected, decreases the level of MPs identified within tissue samples depending on the approach used (Table 1). After blank subtraction adjustments, the total MPs identified within tissue samples have a mean of 14.99± 17.18 MP/g of tissue. Four vein tissue samples (samples 1, 2, 4, 5) fit the criteria for using a LOD and LOQ calculation, showing 4.94–22.22 MP/g for alkyd resin, 5.56–52.17 MP/g for PVAc, 4.35–7.95 MP/g for nylon/EVOH EVA tie layer, 5.56 MP/g for PVAE, and 5.24 MP/g for PUR, above the quantification threshold.

## Discussion

This report provides evidence of MPs within human vein tissue samples, using a robust, best practice, background contamination approach with μFTIR chemical composition analysis to verify the particles present. In total, 20 MPs were identified within 4 of the 5 vein tissue samples, with an unadjusted average of 29.28 ± 34.88 MP/g of tissue. By subtracting any MPs detected in the corresponding blanks (without differentiating polymer types), an adjusted mean of 14.99 ± 17.18 MP/g tissue sample is reported. The MP levels within tissue samples were not significantly higher ($p$ = 0.293) than those identified within combined procedural blanks, although, as stated, the MP polymer types detected in the vein samples were different to those detected in the background control samples (10.4 ± 9.21 MP/sample).

The levels of MPs in the vein tissues are similar to those reported for colon (28.1 ± 15.4 MP/g of tissue) [13] and higher than those reported for lung (3.12 ± 1.3 MP/g of tissue) [12]. The background contamination compares similarly to previous investigation of MPs in lung from surgery patients, where the same digest, filtration, µFTIR approach with procedural blanks were employed [12]. The lung study reported a mean combined non-MP procedural blank results 9.04 ± 4.84 non-MP particles per sample [12]. In contrast, no particles were reported in the controls used in the colon study where similar digestion, filtration and µFTIR plus SEM/EDX analyses was employed on a subset of particles from samples [13]. In terms of the predominant shape, fragment was more common in vein and lung samples while colon tissue samples were 96% fibres [12,13]. Comparing the means sizes of the MP particles: lung samples (measured using the same Omnic Picta software tools) had a mean particle length of 223.10 ± 436.16 µm (range 12–2475 µm), and a mean particle width of 22.21 ± 20.32 µm (range 4–88 µm), the colon study did not report sizes, the vein samples were similar lengths with a mean length of 119.59 ± 226.82 µm (range 16–1074 µm), and a larger mean particle width of 41.27 ± 62.80 µm (range 7–300 µm), which, could be hypothesised, reflects the logistics of a particle entering increasingly smaller airway channels compared with vein diameters.

Of the MPs detected, five polymer types were identified with alkyd resin, PVAc, and nylon/EVOH-EVA, tie layer the most abundant. This contrasts with previous human tissue sample analysis to date, where PP (23%), PET (18%), resin (15%), and PE (10%) were the most abundant for lung (n = 11 samples) [12], and polycarbonate, nylon, and PP for colon (n = 11) [13]. These first human tissue analyses suggest that the distribution of the MP predominant types may be tissue specific. It is important to note that the FTIR spectra for PET and PES (polyester) are similar and can be difficult to distinguish [27,28], however a high match of 70% was adopted to distinguish these MP types in this study.

Of those polymers detected in the vein samples, alkyd resins are used in synthetic paints, varnishes and enamels used for furniture and architectural coatings, product finishes, special-purpose coatings, and car refinishing primers as well as others. Alkyd resins emit solvent vapours and to improve safety against fire and health hazards of organic solvents. their market share has decreased in favour of acrylic, polyurethane (detected in vein sample 2) and epoxy resins [29]. Nylon- and EVOH- EVA tie layer polymers are used to bond plastic polymers to create flexible packaging materials, with blends optimised to improve their characteristics such as helping to prevent moisture intrusion or tensile qualities [30]. Applications include many uses from food packaging and lamination to multi-layer pipe, wire, and cable. PVAc is a co-polymer adhesive, discovered in 1912, also used in food packaging, shipping boxes/bags and binders for paper, plastics and foils. It is one of the main ingredients of wood glue but has a recent biomedical use in DNA/drug delivery [31].

The blank particle MP polymers differed from the vein MP particles observed. A large retrospective study across five hospitals in Massachusetts, U.S.A, identified that the operating theatres contributed to 33.2 tons per year of plastic waste [32]. A study of PET usage in 22 Brisbane operating theatres reported a production ~1,700 kg per year [33]. In our recent study of MP levels and types in a surgery environment, airborne MP levels of 1,924 ± 3,105 MP m$^{-2}$ day$^{-1}$ in theatre were observed [20]. This presents a challenge and introduces limitations for any study that involves obtaining samples from a surgery environment, in minimising background contamination where the patient and surgical procedure are the priority. With respect to the main MP polymers present in a theatre airborne environment, PET was reported as most abundant (38%) followed by PP, nylon, PE, and PTFE (25%, 13%, 8%, 5%, respectively) [20]. This is similar to the profile of MPs observed in the blank samples in this study, with PET>PTFE>PP. None of these MP polymers were detected in the vein samples suggesting that the approach taken by surgeons in placing tissues immediately into glass vials was robust.

The plastic monomer and additives containing particles detected in the vein samples are interesting. Propyleneglycol dilaurate has many uses as an emollient, surfactant, oil-phase ingredient in cosmetics, hair care, creams, lotions; in resinous polymeric food-contact coatings; defoamer in food-contact paper; in food-contact textiles, with the EU E-number E1520. Sorbitan monopalmitate (Span 40) is used as a food additive, specifically an emulsifier, permitted by the EU (as E495). Trimethylolpropane trinonanoate has limited information available in the literature. A closely related chemical, trimethylolpropane triacrylate (TMPTA), is a monomer used for its lubricant and tensile properties, as an additive with natural rubber, ethylene-(vinyl acetate) copolymer (EVA), and rubber/EVA blends [34]. Finally, methyl laurate is a natural chemical derived from coconut oil, also used as a plasticiser polymer and food additive [35]. Of the remaining non-MP particle chemicals most frequently detected: N-methyl-3-piperidylbenzilate HCl is an anticholinergic drug, monoelaidin an activator of transient receptor potential vanilloid subtype 1 (TRPV1), a non-selective cation receptor expressed in perivascular nerve cells as well as vascular endothelial and smooth muscle cells receptor [36–39], and epoxidized soybean oil is used as a plasticizer and stabilizer in polyvinyl chloride (PVC) plastics. The presence of such non-MP polymers suggests that the digestion process has been incomplete, in future work an additional enzymatic digestion for more complete removal of non-MP polymers from relatively fatty human tissues is recommended.

Three further important methodological notes to highlight concern the morphological characteristics of the MP particles, their distribution within the tissues and the apparent presence of a strong $CO_2$ peak in the spectra obtained. For morphology, it was frequently not possible to differentiate between either fragment or film types and the 'irregular' shape category used instead. Such particles displayed a thick film/gel-like consistency, that differed from observations in our previous studies using environmental and biological samples. In terms of tissue distribution, samples were digested as provided, without any dissection to minimise the introduction of background MP contamination. It was, therefore, not possible to identify whether MPs were specifically localised within the adventitial, medial, or intimal layer or were distributed throughout. Finally, the specific μFTIR equipment employed in this study generated $CO_2$ peaks on the sample spectra, which may be attributed to atmospheric $CO_2$ in the room. The equipment used has a sealed desiccant chamber to reduce humidity and the background readings were taken immediately prior to sample spectra readings to reduce interference by atmospheric contaminants, but slight changes in lab conditions may still have been responsible for the peak presence in the absence of a sample chamber that can be purged. Now that MPs have been detected, our future studies will investigate MP distribution in tissue sections and assess the functional impact in relevant cell types.

## Conclusions

This pilot investigation of a small number of human saphenous vein tissues confirms the presence of MPs. The levels observed are similar to, or higher than, those reported in the literature for colon and lung tissues respectively. The size ranges are similar, yet the shape characteristics and polymer types differ from other human tissue types analysed to date. This work is the first to report MP presence in human vascular tissues and provides a starting point for more in-depth analysis of the levels, types, and clinical implications of such presence which will comprise our future work.

## Supporting information

**S1 Table. Showing all identified polymers within vein samples and accounting for the same polymer if identified in controls.** Additional columns present data used to determine

Limit of Detection and Limit of Quantification values. Abbreviations; PVAc, poly (vinyl propi-onate/acetate); PVAE, polyvinyl acetate:Ethylene; PUR, polyurethane.
(DOCX)

## Author Contributions

**Conceptualization:** Jeanette M. Rotchell, Lauren C. Jenner, Emma Chapman, Robert T. Bennett, Mahmoud Loubani, Laura Sadofsky, Timothy M. Palmer.

**Data curation:** Lauren C. Jenner, Robert T. Bennett.

**Formal analysis:** Jeanette M. Rotchell, Lauren C. Jenner, Emma Chapman, Israel Olapeju Bolanle, Mahmoud Loubani.

**Funding acquisition:** Jeanette M. Rotchell.

**Methodology:** Jeanette M. Rotchell, Lauren C. Jenner, Emma Chapman, Robert T. Bennett, Israel Olapeju Bolanle, Mahmoud Loubani, Laura Sadofsky, Timothy M. Palmer.

**Project administration:** Jeanette M. Rotchell, James Hobkirk.

**Resources:** Mahmoud Loubani, Timothy M. Palmer.

**Supervision:** Jeanette M. Rotchell, Mahmoud Loubani, Laura Sadofsky, Timothy M. Palmer.

**Validation:** Jeanette M. Rotchell.

**Visualization:** Jeanette M. Rotchell.

**Writing – original draft:** Jeanette M. Rotchell, Timothy M. Palmer.

**Writing – review & editing:** Jeanette M. Rotchell, Lauren C. Jenner, Emma Chapman, Robert T. Bennett, Mahmoud Loubani, Laura Sadofsky, Timothy M. Palmer.

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
