## [Decision Letter · Decision Letter 0]

12 Sep 2022

PONE-D-22-22648Detection of microplastics in human saphenous vein tissue using μFTIR: a pilot studyPLOS ONE

Dear Dr. Rotchell,

Thank you for submitting your manuscript to PLOS ONE. After careful consideration, we feel that it has merit but does not fully meet PLOS ONE’s publication criteria as it currently stands. Therefore, we invite you to submit a revised version of the manuscript that addresses the points raised during the review process.

We look forward to receiving your revised manuscript.

Kind regards,

Amitava Mukherjee, ME, Ph.D.

Academic Editor

PLOS ONE

Journal Requirements:

2. Please ensure that you have specified (1) whether consent was informed and (2) what type you obtained (for instance, written or verbal, and if verbal, how it was documented and witnessed). If your study included minors, state whether you obtained consent from parents or guardians. If the need for consent was waived by the ethics committee, please include this information.

"This research did not receive any specific grant from funding agencies in the public, commercial, or not-for-profit sectors. It was part funded by a PhD scholarship for LJ in the “Human Health and Emerging Environmental Contaminants” cluster funded by the University of Hull"

"This research did not receive any specific grant from funding agencies in the public, commercial, or not-for-profit sectors. It was part funded by a PhD scholarship for LJ in the “Human Health and Emerging Environmental Contaminants” cluster funded by the University of Hull."

6. Please ensure that you refer to Figure 1 in your text as, if accepted, production will need this reference to link the reader to the figure.

Reviewers' comments:

Reviewer's Responses to Questions

**Comments to the Author**

1. Is the manuscript technically sound, and do the data support the conclusions?

Reviewer #1: Partly

Reviewer #2: Partly

2. Has the statistical analysis been performed appropriately and rigorously? 

Reviewer #1: Yes

Reviewer #2: I Don't Know

3. Have the authors made all data underlying the findings in their manuscript fully available?

Reviewer #1: No

Reviewer #2: Yes

4. Is the manuscript presented in an intelligible fashion and written in standard English?

Reviewer #1: Yes

Reviewer #2: Yes

5. Review Comments to the Author

Reviewer #1: In the scope of the study the authors present an investigation on the analysis of microplastics in human vein tissue. This type of investigation is of high interest for the scientific community and risk assessment. The study was well designed and the authors are using state-of-the-art sampling designs and quality control/quality assurance measures, especially considering blank samples during the surgery process. Afterwards, the samples were digested and analyzed by µ-FTIR microscopy. While instrument was well described, it misses to report important information as requested by Andrade et al. 2020. Further, the spectral matching process and identification of the resulting spectra is hard to follow has the authors only presented the measured spectra but not the assigned spectrum of the reference material. Also, while the authors discuss their results well they miss to discuss the applied method for extraction and identification. As the manuscript was aimed to be a pilot study, such a discussion needs to be included. Further, the spectral assignments cannot be followed as not all data is available (at least for review) and it would be a great addition if these would be provided or uploaded in a data repository. Finally, the presented spectra and the assigned spectra have one thing in common, a strong CO peak. Due to the combination of an oxidative digestion with a direct filtration this is rather striking yet was not addressed properly by a discussion. In total, the manuscript should be revised covering these and the following remarks:

Major comments:

Line 100: The quality control measure should be moved prior to the vein tissue digestion so that the reader can follow the blank process better.

Line 110: The spectrometer section reads partly unsuitable for a scientific publication (too much praising of the system) and misses important information (how was the MCT cooled, which aperture (number) was used during measurements. (see also Andrade et a. 2020). How were particles selected for measurement and particle numbers later calculated? This is currently hampered to be reproduced.

Line 127: How is the match calculated, has this has an high impact on the determined results?

Quality assurance and quality control:

Where the used solutions for digestion be filtered?

Line 175 to 176: A total of 20 MPs does not fit with a maximum number of 32 per sample. Also it should read either 16 +- 14 or 16.00 +- 13.56. Currently the results are hard to follow due to this first sentence.

Table 1: A mean of 10 MPs for the blank cannot be concluded from the other results or the reader is missing information how the results were for calculated after measurement. This is also affecting the values of Table S1.

Line 193 to 194: based on which data (visual image or ir data) was the size measured? This is not clarified in material and methods.

Line 233 to 245: Which techniques did the cited references use to investigate the MPs. As this is a pilot study this should to be discussed.

Line 246 to 253: Same here, which techniques and sample preparation were applied to determine the polymer types in the cited studies.

Line 252 to 253: The authors are using Anodisc, for which reference 27 showed that this impossible for the available size range independent from the maximum match value. Did the OMNIC database contain other polyesters compared to the one provided in reference 27? This would have a far higher likeliness.

Line 278 to 294: The authors performed quite a long term oxidation of the materials to remove the organic matter. How did they assure that all products of this reaction were removed during filtration as filters with a pore size of only 20 nm were applied. Due to the complex chemical nature of the vein tissue, the formation of these materials may also be linked to chemical reactions. See for example Witzig et al. 2020 which showed misidentification issues if applying nitrile rubber gloves. Also, the data showing these entries were not provided in the results and therefore this part cannot be well followed. Please provide full information of all assigned materials which are discussed including the spectrum e.g. via an excel sheet as electronic supporting information.

Any discussion about the digestion efficiency and related issues are missing. Most striking for the reviewer are the common CO peaks of the assigned materials which may indicate issues related to the oxidation in combination with the very fine Anodisc filters applied affecting the polymer composition and should be at least discussed.

Minor comments:

Line 32: nylon-EVA, tie layer is not a proper description of a polymer. Which part of this combination was part of the tie layer? Was this provided in the database?

Line 36: , polytetrafluoroethylene, PTFE > polypropylene, PP > polyethylene

37 terephthalate, PET > polyfumaronitrile:styrene, FNS: This reads awkward and is not fully to understand. What is the propose of > and the abbreviations should be in brackets.

Line 75: What is the propose of the “(with a 5 μm lower size limit of detection)”? It reads very strange here, especially if put into this types of brackets.

Line 124: high spectral resolution: remove high

Line 181: remove the numbers of the blank mean which were just presented in line 178

Line 258 (Polymer Database, 2022): This is a strange type of citation and was not represented in the reference list

Line 396: Remove the yellow highlight

References

Andrade, J.M. et al. (2020) ‘Standardization of the minimum information for publication of infrared-related data when microplastics are characterized’, Marine Pollution Bulletin, 154, p. 111035. Available at: https://doi.org/10.1016/j.marpolbul.2020.111035.

Witzig, C.S. et al. (2020) ‘When Good Intentions Go Bad—False Positive Microplastic Detection Caused by Disposable Gloves’, Environmental Science & Technology, 54(19), pp. 12164–12172. Available at: https://doi.org/10.1021/acs.est.0c03742.

Reviewer #2: Authors have detected MPs types and shapes in human vein tissue samples. This study is a sort of human biomonitoring study.

i) Why the author chose to do with vein tissue NOT alveolar tissue. We can find different types of Nanoplastics in different dimensions and shapes in alveolar tissues.

ii) There are other confounding factors for the deposition of MPs in the vein tissues. The authors have NOT looked into the details of the samples demography and their habits.

iii) Inter laboratory comparison studies would have given nice conformation especially for these kind of studies. B'cos there is proper standard reference materials for MPs.

iv) How the authors confirm that there is NO artifacts in the sample analysis?

v) THe implication of doing such kind of studies is NOT explained in the manuscript

vi) What is the novelty of this studies, because it is only access to human vein tissues and then analysis using microFTIR.

vii) All the images are of poor quality, i think it shows ghost image or artifacts.

viii) Why did NOT the authors try any other conformational analytical studies.

ix) A GC-pyrolysis would have given some knowledge on the distribution of MPs nature

x) The authors could have given internalization of the MPs in the vein tissue.

xi) A graphical abstract would be nice. Putting the information on the possible route of MPs into vein tissues

xii) More sample size would be better with all the detail and background information of the specimen

6. PLOS authors have the option to publish the peer review history of their article (what does this mean?). If published, this will include your full peer review and any attached files.

Reviewer #1: **Yes: **Sebastian Primpke

Reviewer #2: **Yes: **Natarajan Chandrasekaran

---

## [Author Response · Author response to Decision Letter 0]

9 Oct 2022

Dear Dr. Rotchell, Thank you for submitting your manuscript to PLOS ONE. After careful consideration, we feel that it has merit but does not fully meet PLOS ONE’s publication criteria as it currently stands. Therefore, we invite you to submit a revised version of the manuscript that addresses the points raised during the review process. Please submit your revised manuscript by Oct 27 2022 11:59PM. Kind regards, Amitava Mukherjee, ME, Ph.D. Academic Editor, PLOS ONE.

>We would like to thank you for your considered decision, and we are pleased to have an opportunity to undertake the improvement kindly suggested by yourself and the two reviewers. Our response is written following each of the individual editor and reviewers’ comments below, indicated with a “>”. We feel that our manuscript is now improved.

Editor comments:

• Ensure correct manuscript style is followed. 

>We have cross checked with the online guides and modified the manuscript accordingly.

• Please ensure that you have specified (1) whether consent was informed and (2) what type you obtained (for instance, written or verbal, and if verbal, how it was documented and witnessed). If your study included minors, state whether you obtained consent from parents or guardians. If the need for consent was waived by the ethics committee, please include this information.

>The methods section now includes that written consent was obtained and the UK national procedures for obtaining the ethical approval are detailed.

• Please remove Funding information from the Acknowledgement section and update the finding statement if required.

>Funding information now removed from Acknowledgements section.

The current funding statement covers everything and does not need to be changed.

• Add ORCID number.

>Done.

• Remove the additional ethical statement section, these details should be in the methods section.

>Done.

• Please refer to Figure 1 in the manuscript text.

>Added to manuscript text.

Reviewers' comments

Reviewer #1: In the scope of the study the authors present an investigation on the analysis of microplastics in human vein tissue. This type of investigation is of high interest for the scientific community and risk assessment. The study was well designed and the authors are using state-of-the-art sampling designs and quality control/quality assurance measures, especially considering blank samples during the surgery process. Afterwards, the samples were digested and analyzed by µ-FTIR microscopy. 

>We thank the reviewer for highlighting the timeliness and robustness of our investigation.

While instrument was well described, it misses to report important information as requested by Andrade et al. 2020. 

>We thank the reviewer for highlighting this reference from the marine biology microplastics research field and which represents an attempt to standardise FTIR result reporting. We have read the source and used their Table 1 to add the remaining missing pieces of information into the Methods section to comply.

Further, the spectral matching process and identification of the resulting spectra is hard to follow has the authors only presented the measured spectra but not the assigned spectrum of the reference material. Also, while the authors discuss their results well they miss to discuss the applied method for extraction and identification. 

>As above, this detail is now added to the Methods section. We used the 70% match to library spectra criteria (within the Omnic Picta sotware) as is often reported in the literature. As biologists, rather than chemists, we are less able to comment on spectra features in detail and rely on the library search results rather than the visual comparison of spectra. We included a small number of example spectra in Fig 1 because this was well received by the research community in our previously published work. 

As the manuscript was aimed to be a pilot study, such a discussion needs to be included. Further, the spectral assignments cannot be followed as not all data is available (at least for review) and it would be a great addition if these would be provided or uploaded in a data repository. Finally, the presented spectra and the assigned spectra have one thing in common, a strong CO peak. Due to the combination of an oxidative digestion with a direct filtration this is rather striking yet was not addressed properly by a discussion. In total, the manuscript should be revised covering these and the following remarks:

>We have now set up a data repository site and uploaded MP numbers/size data (it is on a 4 month embargo until public but we will change this if/once our manuscript it accepted for publication): https://doi.org/10.6084/m9.figshare.21288060

Since this is new to us, we had only been saving occasional FTIR spectra in the raw data format, usually we cut and pasted a screen shot into a word document to store it alongside a screen shot of the library match results and any images saved of the particle. Now that we know this (and thanks to the reviewer for nudging us this way, you are the first person to raise this), we can improve our data sharing in future studies by saving the raw file data. Please let us know if we set up the figshare correctly, as we said, this is new for us.

>With respect to the strong CO peak, we agree that this is one main feature. With the caveat that we are biologists applying FTIR as an analytical tool, as opposed to analytical chemists with organic chemistry expertise, we understand that the FTIR spectra is taken by the Omnic Picta software and the sharp plus broader peaks, representing dipole bonds are captured. These are compared by the software to a database of library spectra. In that respect, we place our hand on our hearts and openly admit that we trust the software/library search to provide the match. Remember though that we only accept a match at above the 70% similarity threshold. We also analyse a significant number of total particles (~25%) compared to most published studies in this area. To add further confidence, we can state that we see very different MP polymer profiles in different sample types – air samples are very different from biological tissues for instance. The procedural controls also usually display different MP polymers (probably reflecting that airborne source).

Major comments:

Line 100: The quality control measure should be moved prior to the vein tissue digestion so that the reader can follow the blank process better.

>Done as requested.

Line 110: The spectrometer section reads partly unsuitable for a scientific publication (too much praising of the system) and misses important information (how was the MCT cooled, which aperture (number) was used during measurements. (see also Andrade et a. 2020). How were particles selected for measurement and particle numbers later calculated? This is currently hampered to be reproduced.

>Now added this detail to methods section, please see revised draft Lines 150-175. Particles were not pre-selected, all were analysed for one quarter of the filter for each sample.

Line 127: How is the match calculated, has this has an high impact on the determined results?

>This is carried out automatically by the Omnic Picta software in carrying out a search against the libraries used. We added ‘using the Search option’ to make this clearer in the manuscript. Note that we are biologists applying the FTIR technique rather than analytical chemists who use FTIR as their main work, so this is at the limits of our expertise. If a community workshop could be facilitated – could it please be accessible for colleagues with disabilities such as dyscalculia. 

>In summary, our approach is better than the vast number of studies/standard practice seen in the microplastics research community. We will endeavour to keep improving our work based on the reviewer advice.

Quality assurance and quality control:

Were the used solutions for digestion be filtered?

>Yes, this is now made clear in the methods section, apologies for lack of clarity.

Line 175 to 176: A total of 20 MPs does not fit with a maximum number of 32 per sample. Also it should read either 16 +- 14 or 16.00 +- 13.56. Currently the results are hard to follow due to this first sentence.

>Apologies for the confusion, the number 20 is from one quarter area of all the filters combined. Total MPs extrapolated to whole filters would be n=80 but we do not like to make such an assumption, so we state was found. We have changed the first sentence to make this clear. We also changed the values as suggested by the reviewer.

Table 1: A mean of 10 MPs for the blank cannot be concluded from the other results or the reader is missing information how the results were for calculated after measurement. This is also affecting the values of Table S1.

>The blank MP mean value is calculated using the number of MPs found on one quarter of each of the blank filters, multiplied by 4 to arrive at a whole filter, then the mean calculated. 

We have added a sentence to the ‘statistical analysis’ in the methods section to make this clearer.

Line 193 to 194: based on which data (visual image or ir data) was the size measured? This is not clarified in material and methods.

>This detail is now included in the Methods section (Lines 150-152 ). The Omnic Picta software allows such measurements as a visual tool.

Line 233 to 245: Which techniques did the cited references use to investigate the MPs. As this is a pilot study this should to be discussed.

>The techniques used in the studies have been summarised as requested. Finding no MPs in blank samples is highly questionable in our view, but it would be unprofessional/non-collegial of us to state that. 

Line 246 to 253: Same here, which techniques and sample preparation were applied to determine the polymer types in the cited studies.

>The earlier comment has prompted us to add the lung/colon study summaries in approach to the discussion for greater clarity in making comparisons between these studies.

Line 252 to 253: The authors are using Anodisc, for which reference 27 showed that this impossible for the available size range independent from the maximum match value. Did the OMNIC database contain other polyesters compared to the one provided in reference 27? This would have a far higher likeliness.

>We thank the reviewer for highlighting this point. We note that the reviewer is referring to their own published work, which they will obviously know the details better than we will. Omnic Picta uses a suite of libraries which focus on polymers, forensic science chemicals, dyes. It differentiates between PET/PES but, since we are not chemists, we rely on the >70% match to their library spectra as our criteria to decide. Since we have stated this in our methods, and added the sentence to the Discussion that the community doesn’t agree on this by saying it can be “difficult to distinguish”, we respectfully feel we have covered this point as clearly as we can and without misleading any readers.

Line 278 to 294: The authors performed quite a long term oxidation of the materials to remove the organic matter. How did they assure that all products of this reaction were removed during filtration as filters with a pore size of only 20 nm were applied. Due to the complex chemical nature of the vein tissue, the formation of these materials may also be linked to chemical reactions. See for example Witzig et al. 2020 which showed misidentification issues if applying nitrile rubber gloves. Also, the data showing these entries were not provided in the results and therefore this part cannot be well followed. Please provide full information of all assigned materials which are discussed including the spectrum e.g. via an excel sheet as electronic supporting information.

Any discussion about the digestion efficiency and related issues are missing. Most striking for the reviewer are the common CO peaks of the assigned materials which may indicate issues related to the oxidation in combination with the very fine Anodisc filters applied affecting the polymer composition and should be at least discussed.

>We thank the reviewer for these insightful comments. If we understand correctly, there are two points highlighted – the organic matter in veins and doubts around the spectrum obtained.

>In our past research we have analysed MPs in fish (Akoueson et al., 2020), molluscs (Li et al., 2018), human skin (not published), human lung (Jenner et al., 2022), human urine (in prep). This vein work is the most recent. In each study we have adapted based on the previous studies, refining what works well and what does not. The fish samples were often very fatty. That led us to the high volume (200 mL), longer duration (7 d), heated hydrogen peroxide digest. Ultimately, we do still end up analysing a lot of cellulose/cellophane and zein particles. These are included in the total particle count values. We use nitrile gloves in the lab but they never actually come into contact with the sample at any point of the method. In Li et al., 2018, we used individual tweezering onto a diamond platform for FTIR analysis, and arguably there could be contact there, but in this and the lung study we now use the anodisc filters meaning that spectra are obtained directly from the filters with no manual actions required.

>The reviewer is amongst those pioneering the open data approach in the microplastics field, and we applaud this. For us, and most researchers in this field, this is new. Nonetheless, we have reached out to another member of the international community and sought advice on how we make our raw data available. We have made it available here (with a 4 month embargo that we will lift upon publication): https://doi.org/10.6084/m9.figshare.21288060.

Minor comments:

Line 32: nylon-EVA, tie layer is not a proper description of a polymer. Which part of this combination was part of the tie layer? Was this provided in the database?

>We have used the exact description as it was categorised/reported by the Omnic Picta database. It is literally the words they use. We always transcript the exact wording from the database matches.

Line 36: , polytetrafluoroethylene, PTFE > polypropylene, PP > polyethylene

37 terephthalate, PET > polyfumaronitrile:styrene, FNS: This reads awkward and is not fully to understand. What is the propose of > and the abbreviations should be in brackets.

>Modified as suggested by the reviewer.

Line 75: What is the propose of the “(with a 5 μm lower size limit of detection)”? It reads very strange here, especially if put into this types of brackets.

>Deleted as suggested by the reviewer.

Line 124: high spectral resolution: remove high

>Done.

Line 181: remove the numbers of the blank mean which were just presented in line 178

>Done.

Line 258 (Polymer Database, 2022): This is a strange type of citation and was not represented in the reference list

>Apologies for this omission in the reference list. It is a website and we have now added the URL as the reference source.

Line 396: Remove the yellow highlight

>Done.

References

Andrade, J.M. et al. (2020) ‘Standardization of the minimum information for publication of infrared-related data when microplastics are characterized’, Marine Pollution Bulletin, 154, p. 111035. Available at: https://doi.org/10.1016/j.marpolbul.2020.111035.

Witzig, C.S. et al. (2020) ‘When Good Intentions Go Bad—False Positive Microplastic Detection Caused by Disposable Gloves’, Environmental Science & Technology, 54(19), pp. 12164–12172. Available at: https://doi.org/10.1021/acs.est.0c03742.

>Thank you.

Reviewer #2: Authors have detected MPs types and shapes in human vein tissue samples. This study is a sort of human biomonitoring study.

i) Why the author chose to do with vein tissue NOT alveolar tissue. We can find different types of Nanoplastics in different dimensions and shapes in alveolar tissues.

>We had access to such tissue via collaboration with surgeons and since virtually nothing is known about the presence and types of MPs in humans, we considered this an opportunity to start somewhere. The cardiovascular surgeons are interested in the data for several reasons as outlined in the Introduction section. Nanoplastics are beyond the scope of this work and has very different implications compared with the microplastics range. The two cannot really be compared in that the ways to detect them are very different and their likely impacts probably very different too. We focus on MPs only and as such are doing novel work, which is kindly highlighted above by Reviewer 1.

ii) There are other confounding factors for the deposition of MPs in the vein tissues. The authors have NOT looked into the details of the samples demography and their habits.

>We agree with the reviewer in that such information would be ideal. The terms of the ethical consent we obtained, however, did not allow this. We could only use the metrics we include in the Table. In any larger scale study, provided we get this published/find funding in a dire Brexit landscape, we would include 10s more patients from diverse groups and would seek ethical consent for such information.

iii) Inter laboratory comparison studies would have given nice conformation especially for these kind of studies. B'cos there is proper standard reference materials for MPs.

>We agree and will seek such collaborations for a full scale study. Indeed, we are now working with a colleague in The Netherlands who has agreed to supply reference samples of PP and nylon fibres for future studies.

iv) How the authors confirm that there is NO artifacts in the sample analysis?

>The procedural blanks quantify and characterise the background particles present and are documented in the results section. The FTIR match (of more than 70% acceptance level) removes much of the uncertainty around a particle being an artifact. The colon study cited (Ibrahim et al., 2021) state that they find no MPs in the procedural blank samples – that is more worrying in our opinion. Given that we state our methods (more clearly thanks to reviewer 1’s comments), we are confident that any journal reader can make up their own mind on whether particles we find are the polymer that the FTIR microscopy/library spectra database search reports, or an artifact.

v) THe implication of doing such kind of studies is NOT explained in the manuscript

vi) What is the novelty of this studies, because it is only access to human vein tissues and then analysis using microFTIR.

>Responding to comments v and vi together: The work is novel, as reviewer 1 has highlighted, in that MPs are only recently being detected in human tissues: lung, blood, colon and herein vein. The introduction explains the rationale from the point of view of the surgeons involved as co-authors/investigators in the study too – they are interested in vascular health impacts, be they inflammation/oxidative stress or other possible impacts.

vii) All the images are of poor quality, i think it shows ghost image or artifacts.

>The images are taken using the microFTIR microscope software package – there is no way to make better images. This is what you see when doing the analysis. A spectra is obtained from the particle viewed and it is this that is then matched to a library of spectra. We are not identifying these particles based on the visual images; we use the spectra match. You can only use these images to show shape (eg, fibre or fragment). We do not use the images to measure the length/width either – instead you can draw a box around the particle and measure the dimensions more accurately in that manner. These are methods within the Omnic Picta Software package and we are careful to cite that we use this package in our methods section.

viii) Why did NOT the authors try any other conformational analytical studies.

>This is a good suggestion and, provided we receive funding for a larger scale study, we would add electron microscopy analysis in parallel. Please note that we do say this is a pilot analysis and have kept the manuscript short/concise in accordance with this.

ix) A GC-pyrolysis would have given some knowledge on the distribution of MPs nature

>There’s pros and cons of the different methods and we take this comment on board for future work. 

x) The authors could have given internalization of the MPs in the vein tissue.

>The scope of this work was to detect if any MPs were present, and if there were, characterise them. This keeps the preliminary study focussed and concise. We have plans to micro dissect vein layers and do separate digests/analysis if we can obtain funding.

xi) A graphical abstract would be nice. Putting the information on the possible route of MPs into vein tissues

>We agree that a graphical abstract would be nice and have now added one. Suggesting routes into the veins is beyond the scope of this work so we have focussed on an image like our lung MP manuscript (Jenner et al., 2022), which was well received in the media.

xii) More sample size would be better with all the detail and background information of the specimen

>There is a trade off in that we analyse what the surgeons provide. In surgery, the surgeons are focussed on the patient’s health/operation and not on maximising larger sizes of tissue for our analysis. We feel this is the correct hierarchy of decision making. The ethical consent in place prohibited use of data beyond that which we state in Table 1.

6. PLOS authors have the option to publish the peer review history of their article (what does this mean?). If published, this will include your full peer review and any attached files.

Do you want your identity to be public for this peer review? For information about this choice, including consent withdrawal, please see our Privacy Policy.

Reviewer #1: Yes: Sebastian Primpke

Reviewer #2: Yes: Natarajan Chandrasekaran

>Again, we thank the two reviewers for their comments. We feel that you have helped us improve the manuscript and you have done so in a kind, constructive way.

>We have now uploaded the Fig 1 to PACE and created a TIF file format.

---

## [Decision Letter · Decision Letter 1]

9 Nov 2022

PONE-D-22-22648R1Detection of microplastics in human saphenous vein tissue using μFTIR: a pilot study

PLOS ONE

Dear Dr. Rotchell,

Thank you for submitting your manuscript to PLOS ONE. After careful consideration, we feel that it has merit but does not fully meet PLOS ONE’s publication criteria as it currently stands. Therefore, we invite you to submit a revised version of the manuscript that addresses the points raised during the review process.

We look forward to receiving your revised manuscript.

Kind regards,

Amitava Mukherjee, ME, Ph.D.

Academic Editor

PLOS ONE

Reviewers' comments:

Reviewer's Responses to Questions

**Comments to the Author**

1. If the authors have adequately addressed your comments raised in a previous round of review and you feel that this manuscript is now acceptable for publication, you may indicate that here to bypass the “Comments to the Author” section, enter your conflict of interest statement in the “Confidential to Editor” section, and submit your "Accept" recommendation.

Reviewer #1: (No Response)

Reviewer #2: All comments have been addressed

2. Is the manuscript technically sound, and do the data support the conclusions?

Reviewer #1: Partly

Reviewer #2: Partly

3. Has the statistical analysis been performed appropriately and rigorously? 

Reviewer #1: Yes

Reviewer #2: I Don't Know

4. Have the authors made all data underlying the findings in their manuscript fully available?

Reviewer #1: No

Reviewer #2: Yes

5. Is the manuscript presented in an intelligible fashion and written in standard English?

Reviewer #1: Yes

Reviewer #2: Yes

6. Review Comments to the Author

Reviewer #1: The authors addressed most of the remarks in appropriate manner. Still a few remarks remain and had not been fully addressed by the authors.

Major issues:

The authors did not include any assigned reference spectra of the Omnic Picta software to the spectra shown in Figure 1. For the reader the result cannot be followed in an easy manner for non-FTIR experts. This will even increase the impact of their manuscript.

Further, the authors state in their answers to the reviewer that the raw data of the FTIR spectra or measurement data was not saved or stored. Maybe I am misunderstood this part, but this is rather unusual for the various systems known to the reviewer, that the measurment files and related data were not stored. The reviewer acknowledges that the authors will consider a better data storage within their future work. Still, due to the embargo on the data on figshare, it was not possible to assess it for review to access it nor was it available as electronic supporting material for review only.

Regarding the strong CO peak:

This is an important feature of all assigned spectra, so it also should be shortly highlighted in the discussion. It is still missing but will be important for future readers to contextualize the results.

Minor Issues:

Material and methods for FTIR: The current text is still not matching the recommended details of the Andrade et al. 2020 paper which can be found in the conclusions.

Reviewer #2: An inter lab study would suffice for the data validation

FT Raman analysis to confirm microplastics in human saphenous vein tissue

7. PLOS authors have the option to publish the peer review history of their article (what does this mean?). If published, this will include your full peer review and any attached files.

Reviewer #1: **Yes: **Sebastian Primpke

Reviewer #2: No

---

## [Author Response · Author response to Decision Letter 1]

25 Nov 2022

We have uploaded a "response to reviewers comments" document.

---

## [Decision Letter · Decision Letter 2]

4 Jan 2023

Detection of microplastics in human saphenous vein tissue using μFTIR: a pilot study

PONE-D-22-22648R2

Dear Dr. Rotchell,

We’re pleased to inform you that your manuscript has been judged scientifically suitable for publication and will be formally accepted for publication once it meets all outstanding technical requirements.

Kind regards,

Amitava Mukherjee, ME, Ph.D.

Academic Editor

PLOS ONE

Additional Editor Comments (optional):

Reviewers' comments:

Reviewer's Responses to Questions

**Comments to the Author**

1. If the authors have adequately addressed your comments raised in a previous round of review and you feel that this manuscript is now acceptable for publication, you may indicate that here to bypass the “Comments to the Author” section, enter your conflict of interest statement in the “Confidential to Editor” section, and submit your "Accept" recommendation.

Reviewer #1: All comments have been addressed

2. Is the manuscript technically sound, and do the data support the conclusions?

Reviewer #1: Yes

3. Has the statistical analysis been performed appropriately and rigorously? 

Reviewer #1: Yes

4. Have the authors made all data underlying the findings in their manuscript fully available?

Reviewer #1: Yes

5. Is the manuscript presented in an intelligible fashion and written in standard English?

Reviewer #1: Yes

6. Review Comments to the Author

Reviewer #1: Thank you for addressing all my concerns and clarifying the process of data storage for their FTIR measurements. Still, reporting the raw spectra (even in the .psa format) for data scientist to develop improved or novel tools for data analysis.

7. PLOS authors have the option to publish the peer review history of their article (what does this mean?). If published, this will include your full peer review and any attached files.

Reviewer #1: **Yes: **Sebastian Primpke

---

## [Editor Report · Acceptance letter]

6 Jan 2023

PONE-D-22-22648R2 

Detection of microplastics in human saphenous vein tissue using μFTIR: a pilot study 

Dear Dr. Rotchell:

I'm pleased to inform you that your manuscript has been deemed suitable for publication in PLOS ONE. Congratulations! Your manuscript is now with our production department. 

Kind regards, 

on behalf of

Professor Dr. Amitava Mukherjee 

Academic Editor

PLOS ONE